# Insights from the transcriptome and metabolome into the molecular basis of diapause in *Leguminivora glycinivorella* (Lepidoptera, Olethreutidae)

Bingshuo Shi[1,2], Bei Chen[1,2], Zhentao Ren[3], Haimeng Zhao[1,2], Laipan Liu[3], Biao Liu[3], Kun Xue[1,2]*

1 Key Laboratory of Ecological Environment in Minority Areas, State Ethnic Affairs Commission, Minzu University of China, Beijing, China, 2 College of Life and Environmental Sciences, Minzu University of China, Beijing, China, 3 Nanjing Institute of Environmental Sciences, Ministry of Ecology and Environment, Nanjing, China

* xuekun@muc.edu.cn

## Abstract

*Leguminivora glycinivorella* is a major pest in soybean production, causing huge economic losses to soybean production when the damage is severe, It typically usually overwintering with mature larvae lagging, which may be an effective tool for pest management. In this study, the key substances of diapause regulation were identified by differential expression genes and differential metabolites, using transcriptome and metabolomics analyses in the diapause and pre-diapause stages in *L. glycinivorella*. The findings revealed that 5558 genes in total were significantly altered during diapause, with pyruvate kinase (PK), trehalose synthase (TPS), superoxide dismutase (SOD), citrate synthase (CS), and 20-hydroxyecdysone (20E) showing significant decreases. There were 1628 metabolites with significant changes, especially proline, Phosphatidyl choline, and unsaturated fatty acids increased significantly in the period of diapause. Meanwhile, KEGG analysis based on the above differential substances showed that they were involved in TCA cycle, glycolysis, and glycerophospholipid metabolic pathway, which suggesting that they may be closely related to energy reserve, antioxidant regulation and hormone regulation during diapause. In this study, we present a comprehensive transcriptomic and metabolomic analysis that identifies three important molecular events during diapause (energy reserve, immune enhancement, hormone regulation) that may play a role in survival and stress resistance during diapause. These findings have greatly improved our understanding of diapause of *L. glycinivorella*, provided a theoretical basis for clarifying the molecular mechanism of diapause regulation of *L. glycinivorella*, and further improved the level of prediction and comprehensive management of *L. glycinivorella*.

**Data availability statement:** The raw RNA-seq data are freely available in the NCBI database under accession no. PRJNA1246935. Other data are within the paper and its Supporting Information files.

**Funding:** This work was funded by Minzu University of China. Grant Number is KLEEMA0202102 and Grant Recipient is KX. The funders had no role in study design, data collection and analysis, decision to publish, or preparation of the manuscript.

**Competing interests:** The authors have declared that no competing interests exist.

# 1 Introduction

*Leguminivora glycinivorella* Mats, commonly known as the soybean pod borer (SPB), belongs to the insect class Lepidoptera, family Olethreutidae, and genus Leguminivora. SPB is the most important soybean pest, posing a major threat to soybean production throughout northeastern Asia [1–3]. SPB eat seeds and burrow their larvae into pods, which stunts soybean growth. In China, the pest feeding rate on soybean is 10% to 30% in general, and even to more than 50% in the year with pest outbreaks. However, it is difficult to control SPB with pesticide [2,4]. Chemical control of soybean pests is the main method, but long-term use can easily lead to pesticide residues and pest resistance, which will affect human health and ecological security. This pest overwinters with mature larvae, and by preventing them from entering diapause or exiting diapause, could force the insect to be active under harsh climatic conditions and evoke ecological suicide [5,6]. Therefore, Studies on the diapause of soybean pod borer are important for controlling pest population outbreaks.

Diapause is defined as a period of suspended development in insects and other invertebrates during unfavorable environmental conditions [7]. In insects, diapause is a means of avoiding mortality and adjusting to the harsh surroundings [8]. Metabolic suppression is a diapause mark that enables insects to synchronize their life cycles to coincide with times that are suitable for development, growth, and reproduction and survive unfavorable seasons [9]. Diapause is a developmental stop state that is genetically determined and can happen at the embryonic stage [10], larval stage [11], pupal stage [12] and adult stage [13]. Diapause types are divided into obligate and facultative according to whether it can be influenced by environmental factors [14]. Insect diapause is a dynamic process successive phases, which includes three periods: pre-diapause, diapause and post-diapause. In the stage of pre-diapause, SPB stores food, changes behavior and reduces development when the stimuli reach some critical level [15]. Insect diapause is characterized by changes in the body, primarily a decreased metabolic rate and an improved tolerance to temperature and moisture [16,17]. When the conditions change to be suitable for SPB, the insects recover from diapause with the metabolic and the physiological level returning to normal [6]. SPB is a typical univoltine insect with obligatory diapause, and the larvae enter the pods to feed and develop until they are fourth instar larvae. They leave the pods, and overwinter in cocoons in the soil layer of 3–15 cm [18–20]. In order to achieve the goal of controlling the reproduction of this pest in a more environmentally friendly way, it is necessary to understand the molecular mechanism of diapause in SPB.

Previous studies on diapause of SPB include regulation of diapause related metabolites (lipids, sugars), enzymes, genes and hormones [21–25]. For example, studies have shown lipid, total sugar, trehalose, glycogen content and water content of SPB changed significantly before and after diapause [22]. The adjusting of those biochemicals, including trehalose and glucose, was found as an adaptation to low temperature [23]. In addition to external factors such as temperature, diapause in SPB may also be regulated by hormones in their bodies. The contents of ecdysone,

juvenile hormone, prothoracicotropic hormone and cytochrome P450 in diapause SPB are significantly different from those in the stages of non-diapause. The contents of ecdysone, juvenile hormone, and cytochrome P450 in worms during the diapause showed a decreasing and then increasing trend, and the lowest content appeared in February, indicating that February may be the critical period from diapause to non-diapause of SPB [24]. Recent studies comparing diapause larvae with mature larvae, and mature pupae with newly developed pupae using transcriptomic and proteomic, and FAS and small heat shock protein genes have been identified as playing key roles in diapause regulation and larval survival [25]. In summary, the molecular mechanisms related to diapause in SPBs are still in a preliminary state.

Metabolomics is the subject of qualitative and quantitative analysis of small molecules simultaneously, mainly studying the exposure of living organisms Changes of metabolites in vivo after boundary stimulation may change regularly with time [26]. Transcriptomics (RNA-Seq) is an important method to study gene function and structure. Studying various genes in individuals, tissues or cells under different conditions is an important aspect of genomics, which has short sequencing time, large sequencing capacity and low sequencing cost [27,28]. The combination of the transcriptome and metabolome better reflects the environment of the cell. The aims of this research are analyzing differentially expressed genes and metabolites and their functions of SPB in diapause and pre-diapause period with RNA-seq technology and targeted liquid chromatography mass spectrometry (LC-MS), and to elucidate the molecular mechanisms underlying diapause.

## 2 Materials and methods

### 2.1 Test insect

All samples were collected from the soybean fields in Xing'an League, Inner Mongolia (122°52'E, 46°49'N) on September, 2023. The forth instar larvae, pre-diapause (PD, were white and still feeding) and diapause (D, were red and had stopped feeding and developing for about 30d, S1 Fig), were stored in -80°C. A total of 24 samples from 2 periods were collected, and each sample was divided into two equal parts for transcriptomics and metabolomics. (Three replicates per period for the transcriptome and nine replicates per period for the metabolome).

### 2.2 RNA extraction and RNA-Seq

Total RNA was isolated from the tissue specimens employing TRIzol® Reagent in strict adherence to the supplier's protocol. Subsequently, RNA integrity assessment was conducted using an Agilent 5300 Bioanalyser system, followed by spectrophotometric quantification with a NanoDrop ND-2000 instrument. At Shanghai Majorbio Bio-pharm Biotechnology Co., Ltd. (Shanghai, China), the RNA purification, reverse transcription, library building, and sequencing procedures were completed. The workflow comprised: PolyA + mRNA enrichment via oligo (dT) magnetic bead selection followed by chemical fragmentation; First-strand cDNA synthesis with random hexamer primers using SuperScript™ ds-cDNA Synthesis Kit (Invitrogen, CA); Second-strand synthesis, end-repair/A-tailing, and Illumina adapter ligation per standardized workflows; Size selection (~300 bp fragments) through 2% Low Range Ultra Agarose electrophoresis; PCR amplification (15 cycles) with Phusion High-Fidelity DNA Polymerase (NEB, USA); Library quantification via Qubit® 4.0 Fluorometer prior to paired-end sequencing (PE150) on NovaSeq 6000 Plus platform with NovaSeq Reagent Kits. Clean reads in orientation mode were individually aligned to the reference genome using HISAT2 software [29]. http://www.ncbi.nlm.nih.gov/datasets/genome/GCF_023078275.1/ is the website from which we obtained the SPB reference genome. Using a reference-based process, StringTie put together each sample's mapped reads [30]. Each transcript's expression level was determined using the transcripts per million reads (TPM) method. The analysis of differential expression was done with DESeq2 [31]. Genes with FDR = "0.05" and |log2FC| ≧ 1 were identified as substantially divergent expressed genes (DEGs) [32]. GO and KEGG functional-enrichment analysis were carried out by Goatools and Python scipy software, respectively, were performed to identify DEGs that were significantly enriched in GO and metabolic pathways.

## 2.3 Extraction and analysis of metabolite

50 mg SPB were transferred into 2 mL cryotubes preloaded with 6 mm stainless steel grinding beads. Metabolite extraction was initiated by adding 400 μL of ice-cold methanol:water (4:1, *v/v*) solution containing 0.02 mg/mL L-2-chlorophenylalanine (internal standard). Mechanical homogenization was performed using a Wonbio-96c cryogenic grinder (Shanghai Wanbo Biotechnology Co., Ltd.) at 50 Hz for 6 minutes under -10°C conditions, immediately succeeded by ultrasonication at 40 kHz for 30 minutes in a temperature-controlled chamber (5°C). Following this, the homogenates underwent: ① Cryopreservation at -20°C for 30 minutes to enhance macromolecule precipitation; ② Phase separation via centrifugation at 13,000×g for 15 minutes (4°C); ③ Precise supernatant collection using calibrated micropipettes; ④ Final filtration through 0.22 μm into LC-MS certified vials prior to instrumental analysis.

The SPBs were subjected to LC-MS/MS analysis at Majorbio Bio-Pharm Technology Co. Ltd. (Shanghai, China) using a UHPLC-Q Exactive HF-X system, equipped with an ACQUITY HSS T3 column (100 mm × 2.1 mm i.d., 1.8 μm; Waters, USA). The mobile phases consisted of 0.1% formic acid in water:acetonitrile (95:5, v/v) (solvent A) and 0.1% formic acid in acetonitrile:isopropanol:water (47.5:47.5, v/v) (solvent B). The flow rate was 0.40 mL/min and the column temperature was 40°C. The injection volume was 3 μL.

Thermo UHPLC-Q Exactive HF-X Mass Spectrometer was the mass spectrometer used for data collection. There are two possible modes of operation for its electrospray ionization (ESI) source. The conditions were set as followed: Aux gas heating temperature at 425°C; Capillary temp at 325°C; sheath gas flow rate at 50 psi; Aux gas flow rate at 13 psi; ion-spray voltage floating (ISVF) at -3500V in negative mode and 3500V in positive mode, respectively; Normalized collision energy, 20-40-60 eV rolling for MS/MS. Full MS resolution was 60000, and MS/MS resolution was 7500. Data acquisition was performed with the Data Dependent Acquisition (DDA) mode. The detection was carried out over a mass range of 70–1050 m/z. A uniform format was created from the UHPLC-MS raw data using Progenesis QI software (Waters, Milford, USA) through baseline filtering, peak identification, peak integral, retention time correction, and peak alignment. Then, the data matrix containing sample names, m/z, retention time and peak intensities was exported for further analyses (S1 Table). Majorbio Biotechnology Co., Ltd. (Shanghai, China) self-compiled the Majorbio Database (MJDB), the HMDB (http://www.hmdb.ca/), and Metlin (https://metlin.scripps.edu/) as the primary databases for metabolite identification. Using the R package "ropls" (Version 1.6.2), the discriminant principle component analysis (PCA), least partial squares discriminant analysis (PLS-DA), and a seven-cycle interactive validation were performed [33]. They evaluated the model's stability. Version 1.6.2 of the PLS-DA model's variable importance projection (VIP) was employed to check for differentially accumulating metabolites [34]. Differentially accumulated metabolites (DAMs) were defined as those in which the VIP was greater than 1 and the Fold Change was greater than 2 or less than 0.5. Two groups' distinct metabolites were mapped into their corresponding biochemical pathways using metabolic enrichment and pathway analysis based on the KEGG database (http://www.genome.jp/kegg/).

## 2.4 Combined analysis of the transcriptome and metabolome of SPB

Using internet resources, metabolome and transcriptome data were integrated using OmicsPLS (R packages) software for bidirectional orthogonal projections to latent structures (O2PLS). We were able to extract information about shared pathways for both datasets by connecting DEGs and DAMs to the KEGG pathway database (scipy(Python)).

## 2.5 Quantitative real-time PCR (qRT-PCR) verification

qRT-PCR was used to confirm that the DEG analysis was accurate. Utilizing the TRIZOL Reagent, total RNA was isolated from the two phases previously mentioned. Using the TIANScript RT Kit (Tiangen, Beijing, China), cDNA was created in accordance with the manufacturer's instructions. For real-time qRT-PCR, the QuantStudio3 Real-Time PCR System (Applied Biosystems, Foster City, CA, USA) and SYBR Green kit (Bio-Rad, Hercules, CA) were utilized in accordance with

 

the guidelines provided by the manufacturers. The internal control gene utilized was L. glycinivorella's β-Actin (S2 Table). Melting curve analysis was done to look for primer-dimer production after amplification. Each of the three biological replicates used in the qRT-PCR study has three technical replicates. Using the $2^{-\Delta\Delta CT}$ technique, relative expression levels were determined.

## 3 Results

### 3.1 Transcriptome sequencing

After building six diapause and pre-diapause SPB sequencing libraries, 62.09 gigabytes of clean data were obtained. Strong sequencing results were shown by all samples, with consistently high percentages of bases reaching or above the Q30 criteria (S3 Table). The excellent caliber and dependability of the acquired data are highlighted by these sequencing results, providing a strong basis for further bioinformatics studies and interpretation of the experimental findings. All unigenes were searched across six databases (Nr, Swiss-prot, KEGG, eggNOG, Pfam, and GO) for functional annotation, and 16,731 genes were annotated (S4 Table).

DEGs between the soybean pod borer diapause and pre-diapause groups were found by screening for q < 0.05 and |log2Fold Change| > 1, with 2551 of these being upregulated and 3007 being downregulated, as indicated in S5 Table and S2 Fig. DEGs were categorized into three groups based on their enrichment in the GO database between the diapause and pre-diapause stages of SPB: molecular functions (MF), cellular components (CC), and biological processes (BP). 212 GO terms were created from the up-regulated DEGs (Fig 1A), while 422 GO terms were created from the down-regulated DEGs (Fig 1B). Eighty-four up-regulated and seventy-six down-regulated genes from MF, 1146 up-regulated and

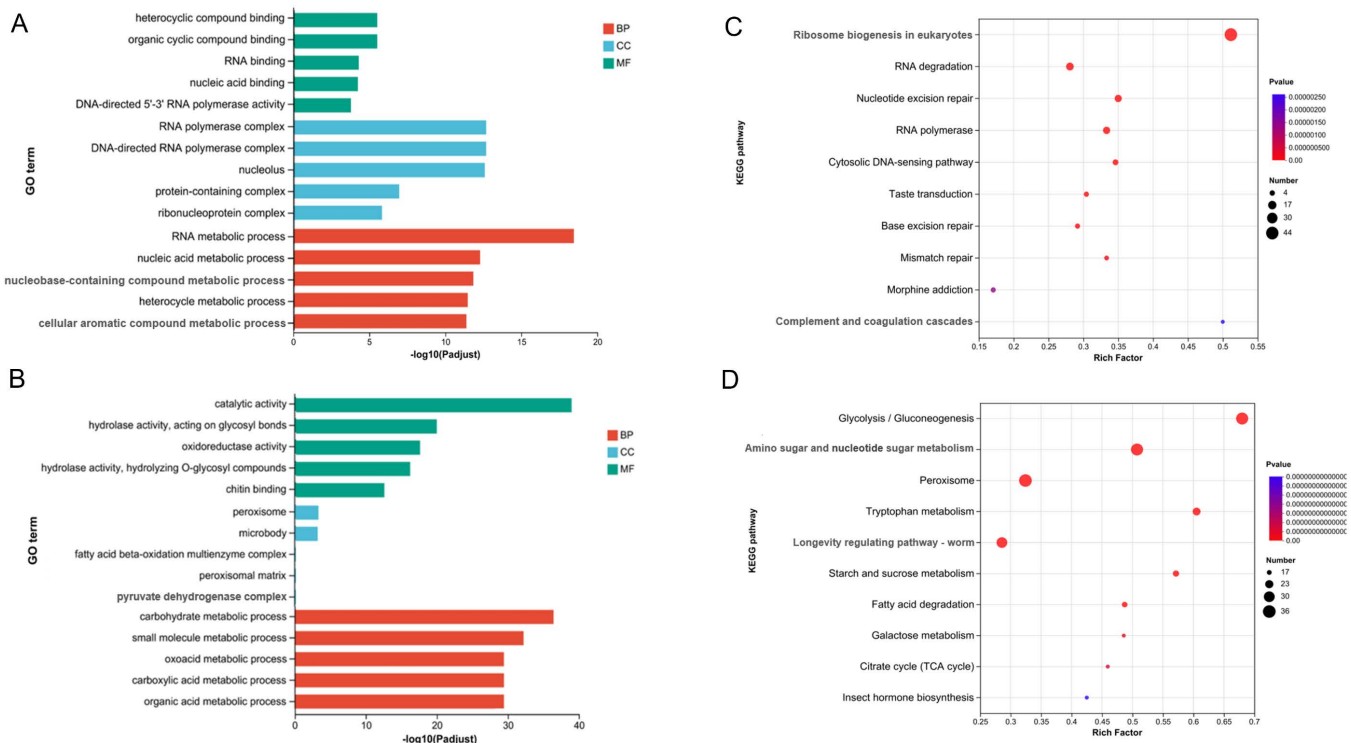

**Fig 1.** (A) Classification of up-regulated DEGs based on GO; (B)Classification of down-regulated DEGs based on GO; (C) KEGG enrichment bubble map of up-regulated DEGs; (D) KEGG enrichment bubble map of down-regulated DEGs.

twenty-three down-regulated genes from BP, and 375 up-regulated and fifteen down-regulated genes from CC were found (S6 Table). According to the GO classification, the DEGs were concentrated in the components of cells and organelles, and they also triggered a number of metabolism and self-regulation processes to withstand adverse environments.

KEGG pathways of DEGs could give deeper insights into the physiological processes related to diapause. Of all gene with a KEGG annotation, 2262 up-regulated DEGs were assigned to 318 KEGG pathways, and 5065 down-regulated DEGs were assigned to 336 KEGG pathways. Up-regulated DEGs were classified into "Ribosome biogenesis in eukaryotes", "RNA degradation", "Nucleotide excision repair", and "RNA polymerase" (Fig 1C). Down-regulated DEGs were classified into "Glycolysis/Gluconeogenesis", "Amino sugar and nucleotide sugar metabolism", "Peroxisome", and "Tryptophan metabolism" (Fig 1D). Top 10 KEGG pathways were significantly enriched, are listed in S7 Table. In addition, some metabolic pathways are also significantly enriched, such as insulin signaling pathway, which may be a major participant in the regulation of metabolites in diaphaesis [35], in which serine/threonine protein kinase (AKT) is involved.

Enrichment with up-regulated differential genes Compared with the KEGG pathway, the number of pathways enriched by down-regulated differential genes was more significant. In the significantly upregulated genes, RPP25L (Ribonuclease P/MRP Subunit P25 Like), RIOK2 (Right Open Reading Frame Kinase 2), IMP3 (IGF2 mRNA-Binding Protein 3) Participation in Ribosome biogenesis in eukaryotes. RPP25L is a ribonuclease complex subunit that repairs damaged rRNA or tRNA and enhances stress resistance under low temperature or oxidative stress [36]. RIOK2 is involved in cell cycle regulation and cell growth [37]. IMP3 belongs to the insulin-like growth factor 2 mRNA binding protein family and usually plays a role in the growth, proliferation and differentiation of cells [38]. It may be related to stress resistance in insect diapause.

Among the down-regulated differential genes, Hexokinase (HK) is involved in Glycolysis/ Gluconeogenesis, amino and nucleotide glucose metabolism, and Galactose metabolism, catalyze the conversion of glucose to glucose 6-phosphate. Pyruvate Kinase (PK) is a key enzyme in glycolysis, and its activity changes affect energy metabolism. 6-phosphofructokinase (PFK) is a kinase One of the major rate-limiting enzymes, catalyzes the production of FDP (1, 6-diphosphate fructose) from Fractose 6-phosphate (F6P), controls the rate of glucose oxidation, and participates in Glycolysis/ Gluconeogenesis. α, α-trehalose-phosphate synthase [UDP-forming] (TPS) is involved in starch and sucrose metabolism and is present in insects. An important enzyme in trehalose metabolism that breaks down trehalose into glucose. In addition, citrate synthase (CS), isocitrate dehydrogenase [NADP] cytoplasmic (IDH) were involved in the citric acid cycle, which were related to the energy metabolism of diapause. Catalase (CAT) and Superoxide Dismutase (SOD) participate in Peroxisome, Longevity regulating pathway-worm. heat shock protein genes (HSP) participate in Longevity regulating pathway-worm. They are closely related to oxidation in diapause larvae. Insect hormone biosynthesis juvenile hormone epoxide in the pathway hydrolase (JHEH) is an important enzyme in the insect hormone biosynthesis pathway, and the downregulation of this gene may cause the accumulation of juvenile hormone levels, which makes insects nutritious in larval state (Other differential genes are shown in S8 Table).

To verify the reliability of the transcriptome sequencing results, six differentially expressed genes were selected for real-time fluorescence quantitative PCR verification, three of which were significantly up-regulated genes from Ribosome biogenesis in eukaryotes. Three significantly down-regulated genes were selected from Peroxisome and Citrate cycle (TCA cycle). Results The expression trend of these 6 differentially expressed genes between pre-diapause group and diapause group was found in transcriptome and real-time fluorescence quantitative PCR. The consistency indicates that the transcriptome sequencing results are reliable (S3 Fig).

## 3.2 Metabolome analyses

Using UHPLC-MS/MS, 1628 metabolites were found in all SPB samples at two physiological stages linked to PD and D. It was determined that these detected metabolites corresponded to nine different biochemical categories (S9 Table). 1628

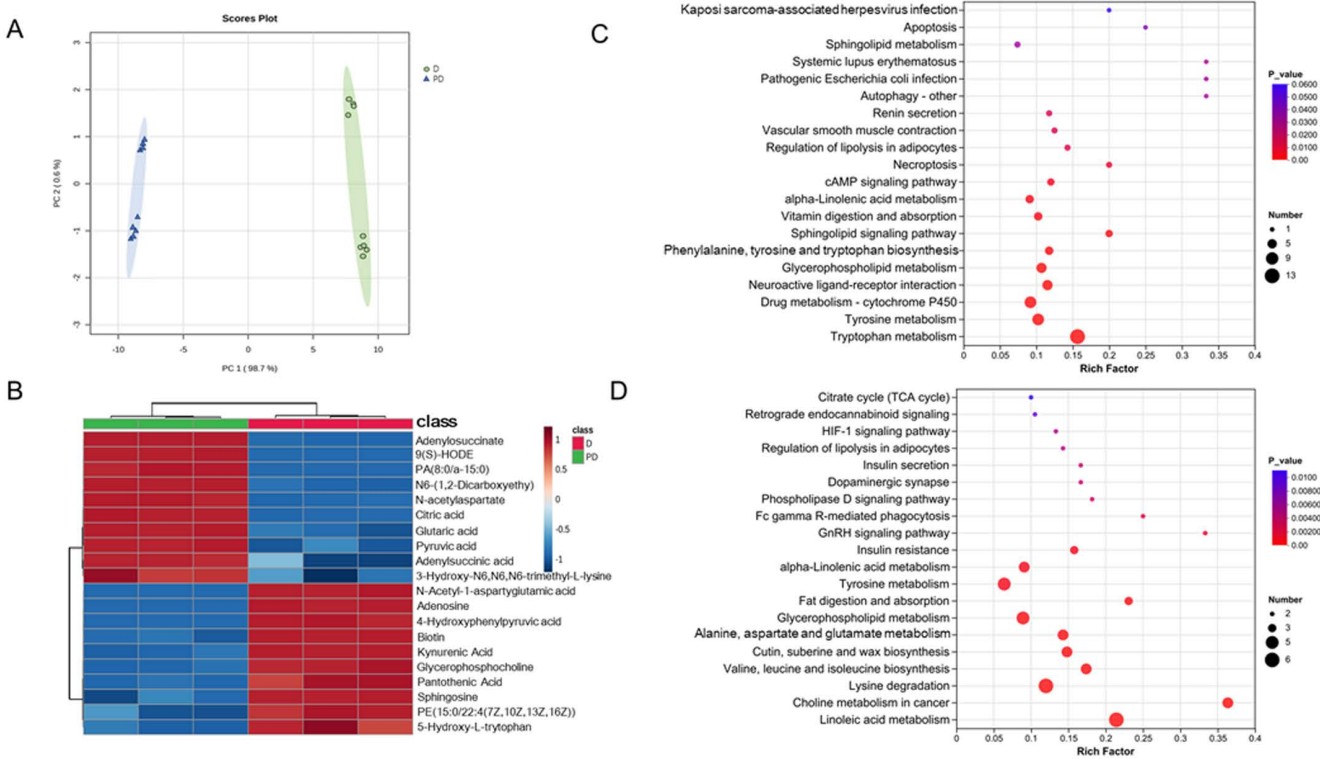

**Fig 2. (A) PCA score scatter plots; (B)Heat map visualization of detected metabolites in PD and D phases; Various colors in the figure represent the relative content.** The metabolite names are shown in the rows, while the samples are represented in the columns; darker shades of blue indicate lower expression levels, whereas darker shades of red indicate higher expression levels; (C) KEGG enrichment bubble map of up-regulated DMs. (C) KEGG enrichment bubble map of down-regulated DMs.

metabolites were subjected to principal component analysis (PCA), which clearly distinguished between D and PD (Fig. 2A). Pairwise comparisons revealed that 107 of these metabolites were differential metabolites. along with a two-way heat map with a hierarchical clustering diagram (Fig 2B) and related data displayed the top ten of these differential metabolites (S10 Table). 50 of 107 were higher in PD (Fig 2B), and the top 5 up-regulated metabolites were Adenylosuccinate, 9(S)-HODE, PA(8:0/a-15:0), N6-(1,2-Dicarboxyethyl)-AMP, and N-acetylaspartate. 57 of 107 were higher in D (Fig 2B), and the top 5 down-regulated metabolites were 5-Hydroxy-L-tryptophan, PE (15:0/22:4(7Z,10Z,13Z,16Z)), Sphingosine, Pantothenic Acid, and Glycerophosphocholine.

To find out the potential metabolic pathways altered between PD and D, KEGG pathway annotation and enrichment analysis were carried out, and the results were displayed in Fig 2C-2D, The top 20 pathways with the most effects on DAMs are shown in S11 Table, The metabolic pathways underpinning the metabolism of amino acids were the ones that were down-regulated in D as opposed to PD. ("Tryptophan metabolism", "Tyrosine metabolism", "Valine, leucine and isoleucine biosynthesis", "Alanine, aspartate and glutamate metabolism") and energy metabolism (i.e., "TCA cycle", "Linoleic acid metabolism", "Glycerophospholipid metabolism"). The up-regulated metabolic pathways in D included "cAMP signaling pathway" and "Sphingolipid signaling pathway". There were revealing that SPB diapause was related to various metabolites, including glycerophospholipid, pyruvate, and arginine, proline.

### 3.3 Combined transcriptomic and metabolomic analyses

Using a thorough transcriptome and metabolomic study, the distinctions between the diapause and pre-diapause SPBs were further clarified. The O2PLS-DA was conducted, and the plots of the O2PLS-DA scores revealed a distinct divergence (Fig 3A). The combined KEGG pathway analysis showed that the changes were enriched in 352 pathways, with 5 pathways linked to the metabolism of amino acids, 3 pathways linked to the metabolism of carbohydrates, and 2 pathways linked to the metabolism of lipids in the top 20 (S12 Table). Glycerophospholipid metabolism, sphingolipid metabolism, and tryptophan metabolism were the most abundant pathways (Fig 3B).

The changes of differential genes and differential metabolites involved in the important pathways of diapause between the prediapause and diapause stages were analyzed, and the metabolic pathways were mapped (Fig 4). There were one DEGs and there DAMs, and five DEGs and two DAMs, seven DEGs and four DAMs, those linked glycerophospholipid metabolism, Glycolysis/Gluconeogenesis metabolism, and Citrate cycle, respectively. The significantly increase metabolites included phosphodimethylethanolamine, glycerophosphocholine, proline, isoleucine, whereas the levels of citric,Phosphatidic acid, pyruvic acid and L-arginine decreased. At the same time, many related genes' transcriptional levels were downregulated, including malate dehydrogenase (LOC125236598, log2FC = −1.93), isocitrate dehydrogenase [NADP] cytoplasmic (LOC125240604, log2FC = −2.03), pyruvate kinase (LOC125242680, log2FC = −1.71), fructose-bisphosphate aldolase (LOC125236660, log2FC = −2.12), pyruvate carboxylaseetc (LOC125235109, log2 FC = −3.10). These results suggested that diapause was related to energy metabolism, including glycerophospholipid metabolism, amino acid metabolism, glycolysis/gluconeogenesis metabolism and citric acid cycle.

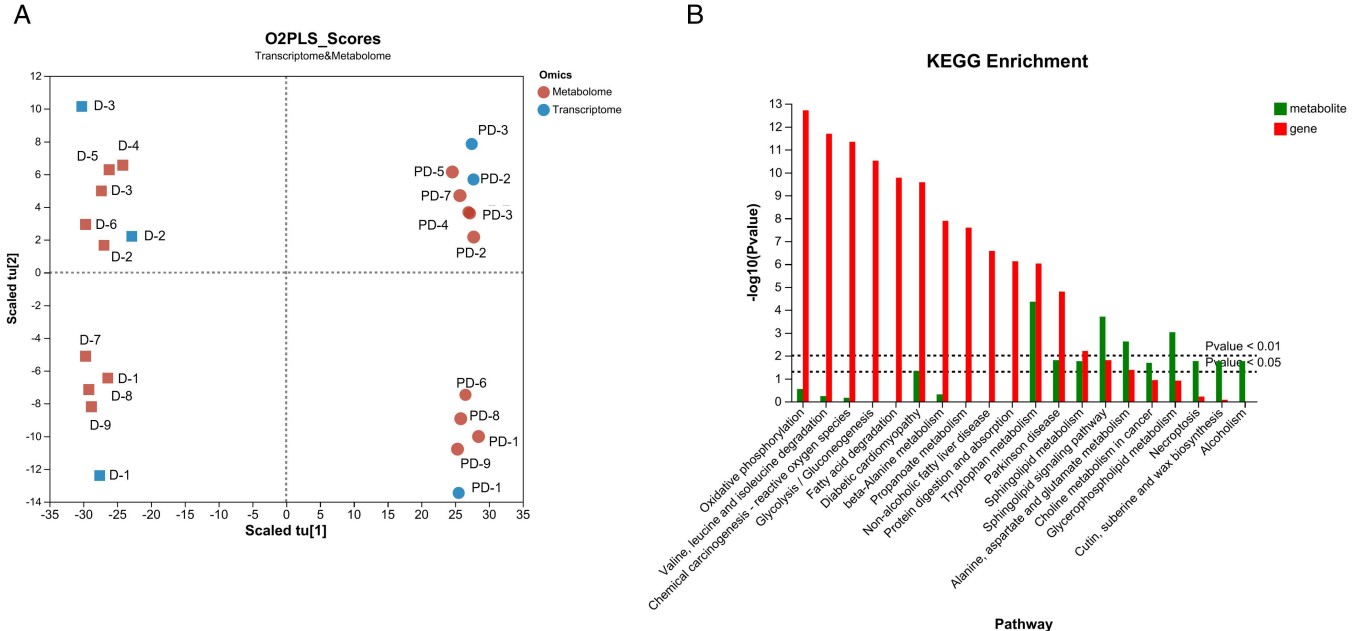

**Fig 3. (A) O2PLS−Scores scatter diagram; (B) Integrated analysis of the top 20 KEGG pathways.**

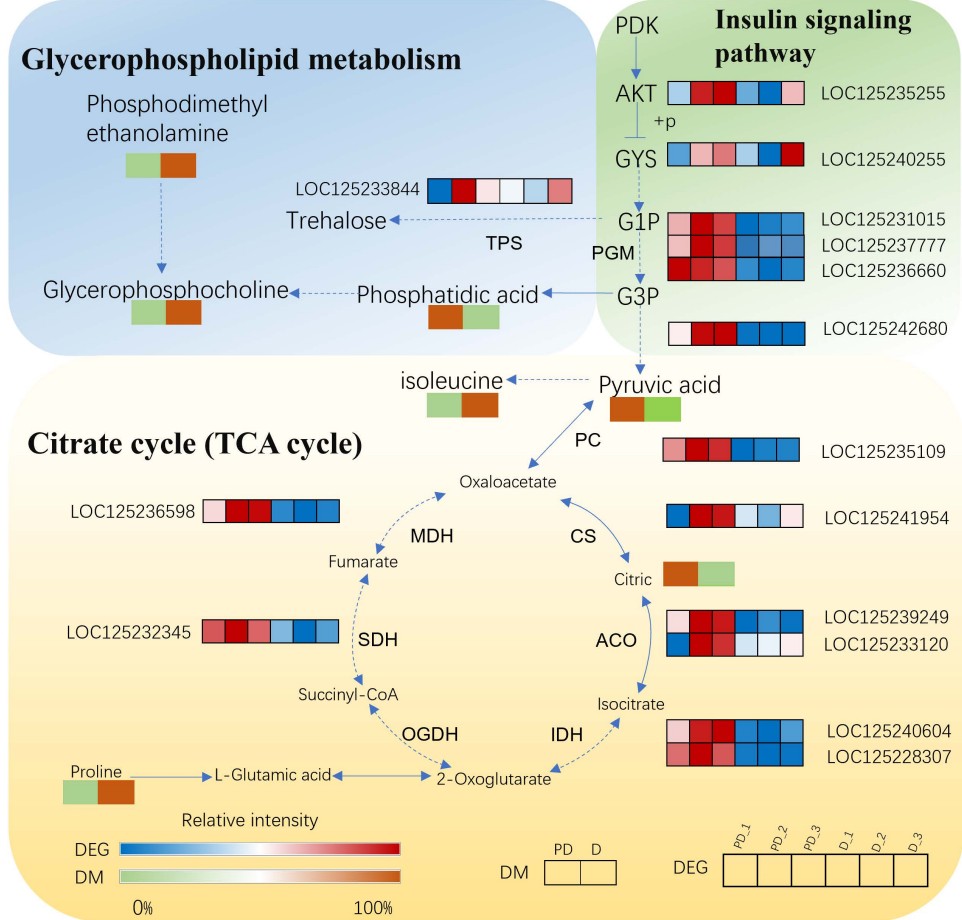

**Fig 4. The profiles of DEGs and DAMs in TCA cycle, insulin signaling pathway and Glycerophospholipid metabolism in response to diapause.**
(PDK: pyruvate dehydrogenase kinase; AKT: RAC serine/threonine-protein kinase; GYS: glycogen [starch] synthase; G1P: Glucose-1-phosphate; G6P: Glucose-6-phosphate; TPS: trehalose-phosphate synthase; PGM: phosphoglucomutase; PC: pyruvate carboxylase; CS: citrate synthase; ACO: cytoplasmic aconitate hydratase; IDH: isocitrate dehydrogenase [NADP] cytoplasmic; OGDH: 2-oxoglutarate dehydrogenase; SDH: succinate dehydrogenase [ubiquinone] cytochrome; MDH: malate dehydrogenase).

## 4  Discussion

### 4.1  Energy storage in the diapause

Diapause strategy is a key determinant for insect survival under seasonal extreme environmental conditions. Considering that soybean insect can regulate its physiological state by timely changing gene expression and metabolite levels to achieve homeostasis, under low temperature stress. Through integrated transcriptomic and metabolomic analyses of SPB, genes associated with diapause were identified. There were three down-regulated DEGs related to insect Antioxidase regulation (heat shock protein genes (HSP), Superoxide Dismutase (SOD), and catalase (CAT)), six down-regulated DEGs related to energy metabolism (hexokinase (HK), pyruvate kinase (PK), ATP-dependent 6-phosphofructokinase (PFK), citrate synthase (CS), isocitrate dehydrogenase [NADP] cytoplasmic (IDH), α, α-Trehalose-phosphate synthase [UDP-forming] (TPS)), and two down-regulated DEGs related to hormone synthesis (juvenile hormone epoxide hydrolase (JHEH) and 20-hydroxyecdysone (20E)). Finding the metabolites associated with diapause-glycerophospholipid, pyruvate, and arginine, proline-laying the foundation for theoretical studies on diapause mechanisms. The diapause method can

provide reference for its prevention and treatment. it is of great significance to study the transcriptome and metabolome of soybean insect at different diapause stages.

The accumulation of energy reserves is a prerequisite for the survival of insects during the diapause period. Almost all insects accumulate nutrients in their bodies [39–43]. Certain metabolites, such as Phosphatidylcholine (PC) and proline, can act as cryoprotectants to improve cold tolerance in diapausing insects [44–46]. Certain transcription factors such as Trehalose, due to its unique chemical properties, is the main sugar circulating in the hemolymph of most insects and plays a critical role in protecting cells from environmental stress [47–49]. trehalose-phosphate synthase (TPS), and trehalose 6-phosphate phosphatase (TPP) were significantly down-regulated and caused trehalose accumulation, which is reflected in the low energy metabolism maintained in SPB. Glycolysis is the most primitive way for cells to obtain energy, and it is also the main pathway in glucose catabolism. As a key enzyme in the first rate-limiting reaction of glycolysis pathway, hexokinase (HK) is widely present in organisms and can catalyze the conversion of glucose to glucose 6-phosphate. 6-phosphofructokinase (PFK) can catalyze the production of FDP from Fractose 6-phosphate (F6P). Its activity can be regulated according to the energy requirements of the cell, controlling the rate of glucose oxidation. In this study, the contents of HK, PK and PFK were significantly reduced, which is presumed to be related to the accumulation of metabolite glucose. The TCA cycle is a common metabolic way to obtain energy in aerobic organisms. In this study, citrate synthase (CS), isocitrate dehydrogenase (IDH) and malate dehydrogenase (MDH) were significantly down-regulated, indicating that the cycle rate of energy supply in insects was significantly lower Metabolic levels are reduced in response to adverse conditions. The metabolite pyruvate, which is an intermediate linking glycolysis and TCA cycles, is significantly down-regulated during diapause, suggesting that the TCA cycle and glycolysis synergistically reduce activity to reduce energy consumption.

Glucose can also be converted into glycerol, which can be used as an anti-freezing protective agent to enhance the cold tolerance of insects [40]. Glycerol kinase (GK) is a rate-limiting enzyme in glycerolipid metabolism by the conversion of glycerol aldehyde 3-phosphate (G3P) to phospholipids. The catabolism of glycerol in vivo consists of a two-step reaction, in which glycerol is catalyzed by glycerol kinase and phosphorylated to produce glycerol 3-phosphate. Rapid cold acclimation of *Sarcophaga crassipalpis* and *Spodoptera exigua* showed that they acted as the main anti-freeze protective substances by increasing their glycerol content [50]. During diapause, GK was significantly down-regulated, This indicates the accumulation of glycerol in SPB, suggesting that this enzyme is mainly used as a protective substance to enhance the cold resistance of insects to survive diapause environment. As a key core factor in insulin signaling pathway, PDK1 phosphorylates and activates serine/threonine protein kinase (AKT), thereby regulating glucose homeostasis [51]. An important gene downstream of AKT is glycogen synthase kinase 3β (GSK3β), which is involved in glycogen biosynthesis and metabolism. GSK3β inhibits glycogen synthase by phosphorylating glycogen synthase (GYS). In this way, it inhibits glycogen biosynthesis [52]. In this study, by inhibiting GSK3β, AKT was able to overcome this inhibition and increase GYS activity, thereby affecting glycogen content. In this study, no significant differences in metabolites such as glycogen, glucose and glycerol were detected, but significant changes in transcription factors were found in this process, conjecturing that the TCA cycle and glycolysis process are dynamic changes, The entire metabolic network is in homeostasis.

Lipid metabolites are closely related to biofilm function. Membrane reconstruction is a common phenomenon during diapause in insects [53]. By increasing the proportion of unsaturated fatty acids in phospholipids and shortening the length of fatty acid chain, the composition of the membrane is adjusted, which is a way for many insects to maintain the stability of membrane fluidity in low temperature environment [54]. Lysophospholipids are produced by removing a fatty acid chain of phospholipids through the hydrolysis of phospholipase, which has a stronger polarity than phospholipids and can produce smaller particles in the water environment. When the conventional cell membrane is exposed to excess lysophospholipids in a state of homeostasis, these foreign lipid substances are integrated with the phospholipid bilayer, and the fluidity of the membrane is also increased [55]. LPA was significantly

up-regulated in SPB during diapause period, which was speculated to be related to the increase of biofilm fluidity, and the improvement of insect resistance and the stability of internal environment. The increase of unsaturated fatty acids, such as 13(S)-HODE, may enhance the membrane binding to membrane proteins, and the increase in unsaturation can cause changes in the physical properties of the membrane, which compensates for the hydrocarbons inside the membrane induced by cold, thereby improving the insect's stress resistance [56]. The combination of results from the metabolome and transcriptome suggests that accumulation of trehalose, proline, PC and unsaturated fatty acids, as well as down-regulation of rate-limiting enzymes in glycolysis and TCA cycling pathways, are the main mechanisms of SPB cold tolerance.

## 4.2 Antioxidant regulation in the diapause

Antioxidant systems enhance animal stress tolerance by assisting animal cells in eliminating excess ROS. The redox potential within the cell can be impacted by hypoxia, which frequently affects diapause insects. Glutathione S-transferase (GST), superoxide dismutase (SOD), catalase (CAT), and peroxidase (POD) are a few enzymatic processes that can be triggered to shield cells from oxidative damage, which is mostly brought on by reactive oxygen species (ROSs) [57]. A large number of studies have shown that the change of protective enzyme activity in insects is closely related to cold resistance. For example, the decrease of SOD activity was also found in the early stage of diapause of *sitodiplosis mosellana* larvae [58]. Catalase (CAT) can eliminate $H_2O_2$ in insects, and CAT plays a major role when $H_2O_2$ concentration is high. During diapause, CAT content decreased significantly, which was speculated to be related to external environmental stress. Superoxide dismutase (SOD), which exists in almost all biological cells, is the most important substance to resist oxidative stress in living organisms, and plays an important role in the physiological process of clearing the high concentration of superoxide free radicals caused by extreme temperature. SOD content was significantly down-regulated during diapause of SPB, which may be related to reducing cell damage.

Heat shock protein (HSP) are highly conserved protein that the body can synthesize in large quantities under stress, this kind of protein is ubiquitous in living organisms. As a molecular chaperone, it assists in the transport and folding of intracellular polypeptides, helps repair and degrade damaged proteins, and plays an important role in maintaining cell survival and internal environment stability [59]. The expression of HSP is closely associated with diapause in insects. In addition to its role as a molecular chaperone, HSP60 also participates in insect immune response and has antioxidant and anti-apoptotic functions. Upregulation of HSP60 is also involved in diapause of goldenrod gall moth [60]. Hsp70 and Hsp90 were upregulated during the onset of diapause in *Locusta migratoria* and *Delia antiqua* [61]. During diapause, many tissues and organs are degraded and rebuilt, and the HSP60 content is significantly up-regulated, which is likely to act as a partner to facilitates the reconstruction of new tissues and organs. These findings suggest that increased antioxidant activity plays a role in the increased cold tolerance during SPB diapause.

## 4.3 Hormonal regulation in the diapause

Juvenile hormone (JH) and ecdysone regulate most life stage transitions in insects, including diapause. Juvenile hormone (JH) is a type of sesquiterpenoid compound synthesized by the pharyngeal body of insects and secreted into hemolymph, which promotes the maturation of gonads, diapause of adult insects and pheromone production in order to maintain the morphology and traits of insect larvae [62]. Juvenile hormone epoxidehydrolase (JHEH) is a juvenile hormone degrading enzyme that plays a key role in the molting process. The decreased expression levels of SmJHE and SmJHEH regulated by diapause in *Sitodiplosis mosellana* were related to the accumulation of JH in the resting stage after diapause, and the decreased expression levels of SmJHE during development might be related to reproductive development [63]. In this study, it was found that JHEH was down-regulated during diapause, thereby inhibiting the breakdown of juvenile

hormones and keeping the larvae in the larval state for a long time, which may be the reason why SPB maintains diapause with mature larvae.

20-hydroxyecdysone (20E) is a steroid hormone synthesized and secreted in the anterior thymus of insects, which plays an important regulatory role in diapause in insect larvae and pupae. Current studies have shown that 20E can promote the synthesis of juvenile hormones [64]. Studies have shown that 20E deficiency leads to decreased ecdysin signaling and juvenile hormone synthesis, which in turn induces diapause for *Colaphellus bowringi* [65]. The study also found that several genes related to molting were differentially expressed after 20E treatment of *silkworm* larvae, such as cuticle protein 3 and chitinase-like protein [66]. In this study, it was found that the content of 20E decreased, and the related chitin synthetase was significantly down-regulated during diapause, which was also consistent with the physiological effect of 20E on promoting insect molting.

## 5 Conclusion

In this study, we report a thorough transcriptome and metabolome analysis that identifies three key biological activities that may contribute to survival and stress tolerance during diapause: energy reserve, antioxidant regulation and hormone modulation. A theoretical foundation for soybean pod borer prevention is provided by these findings. Functional studies on important genes linked to diapause will be carried out in the future using RNA interference technology (RNA interference, or RNAi), with the outcomes being employed in pest management.

## Supporting information

**S1 Fig. The forth instar larvae (PD: pre-diapause; D:diapause).**
(TIF)

**S2 Fig. Volcano plot on differential expression.**
(TIF)

**S3 Fig. Results of fluorescent quantitative verification of DEGs.**
(TIF)

**S1 Table. Identification information of metabolites of *L.glycinivorella.***
(DOCX)

**S2 Table. S2 List of primers used for quantitative real-time PCR.**
(DOCX)

**S3 Table. Sequencing data results of diapause and pre-diapause of *L.glycinivorella.***
(DOCX)

**S4 Table. Summary of the unigenes annotated in different databases for diapause and pre-diapause of *L.glycinivorella.***
(DOCX)

**S5 Table. Differentially expressed genes in the top 10 differences between diapausing phase and pre-diapausing phase of *L.glycinivorella.***
(DOCX)

**S6 Table. The most enriched GO terms of the DEGs between the diapause and pre-diapause of *L.glycinivorella.***
(DOCX)

 

**S7 Table. The top10 enriched KEGG pathway of the DEGs between he diapause and pre-diapause of _L.glycinivorella._**
(DOCX)

**S8 Table. KEGG metabolic pathway and differentially expressed genes in _L.glycinivorella_ during diapause.**
(DOCX)

**S9 Table. General Metabolite Profiles between the diapause and pre-diapause of _L.glycinivorella._**
(DOCX)

**S10 Table. Hierarchical cluster metabolites between the diapause and pre-diapause of _L.glycinivorella._**
(DOCX)

**S11 Table. The top10 enriched KEGG pathway of the DAMs between he diapause and pre-diapause of _L.glycinivorella._**
(DOCX)

**S12 Table the top 20 KEGG pathways enriched in DEGs and DAMs between the diapause and pre-diapause of _L.glycinivorella._**
(DOCX)

## Acknowledgments

We would like to thank the technicians from Shanghai Majorbio Bio-pharm Biotechnology Co., Ltd. for their technical support in the determination of samples. We also appreciate Dr. Munir Ahmad and two anonymous reviewers for their valuable comments and suggestions.

## Author convtributions

**Conceptualization:** Bingshuo Shi, Bei Chen, Haimeng Zhao.

**Funding acquisition:** Kun XUE.

**Investigation:** Bingshuo Shi, Bei Chen, Laipan Liu, Biao Liu.

**Methodology:** Bei Chen, Zhentao Ren.

**Project administration:** Haimeng Zhao, Laipan Liu, Biao Liu.

**Software:** Bingshuo Shi, Biao Liu.

**Supervision:** Zhentao Ren, Laipan Liu, Kun XUE.

**Validation:** Haimeng Zhao.

**Visualization:** Kun XUE.

**Writing – original draft:** Bingshuo Shi.

**Writing – review & editing:** Zhentao Ren.

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
