## [Decision Letter · Decision Letter 0]

11 Feb 2025

PONE-D-24-57514Insights from the transcriptome and metabolome into the molecular basis of diapause in Leguminivora glycinivorella (Lepidoptera, Olethreutidae)PLOS ONE

Dear Dr. XUE,

Thank you for submitting your manuscript to PLOS ONE. After careful consideration, we feel that it has merit but does not fully meet PLOS ONE’s publication criteria as it currently stands. Therefore, we invite you to submit a revised version of the manuscript that addresses the points raised during the review process.

Regarding the reviewers comments, some sections need critical check especially discussion of the results, systematic introduction of the background and relevancy to the study, methodology consistency and improvement of the abstract. 

Please improve it point-wise based on reviewers comments, suggestions etc. Please submit your revised manuscript by Mar 28 2025 11:59PM. If you will need more time than this to complete your revisions, please reply to this message or contact the journal office at plosone@plos.org. Please include the following items when submitting your revised manuscript:

If applicable, we recommend that you deposit your laboratory protocols in protocols.io to enhance the reproducibility of your results. Protocols.io assigns your protocol its own identifier (DOI) so that it can be cited independently in the future. For instructions see: https://journals.plos.org/plosone/s/submission-guidelines#loc-laboratory-protocols. Additionally, PLOS ONE offers an option for publishing peer-reviewed Lab Protocol articles, which describe protocols hosted on protocols.io. Read more information on sharing protocols at https://plos.org/protocols?utm_medium=editorial-email&utm_source=authorletters&utm_campaign=protocols. We look forward to receiving your revised manuscript.

Kind regards,

Munir Ahmad, PhD

Academic Editor

PLOS ONE

Reviewers' comments:

Reviewer's Responses to Questions

**Comments to the Author**

1. Is the manuscript technically sound, and do the data support the conclusions?

Reviewer #1: Partly

Reviewer #2: Yes

2. Has the statistical analysis been performed appropriately and rigorously? 

Reviewer #1: I Don't Know

Reviewer #2: Yes

3. Have the authors made all data underlying the findings in their manuscript fully available?

Reviewer #1: Yes

Reviewer #2: Yes

4. Is the manuscript presented in an intelligible fashion and written in standard English?

Reviewer #1: Yes

Reviewer #2: Yes

5. Review Comments to the Author

Reviewer #1: The authors presented a study aimed at identifying the key substances of diapause regulation by differential expression genes and differential metabolites, using transcriptome and metabolomics in the diapause and pre-diapause stages in L. glycinivorella. They found that metabolic regulation involving TCA cycle, glycolysis, insulin signaling pathway, and glycerophospholipid metabolic pathway, which suggesting that they may be closely related to energy reserve, immune regulation and hormone regulation during diapause. I have a hard time finding key results related to the described discussion… Therefore, I recommend that some revisions be made and specific questions be addressed before the paper is published.

[Abstract]

- In the abstract, insulin signaling is presented as a major pathway; however, there is a lack of related discussion in the results and discussion sections.

[Introduction]

- It is necessary to present a concise and clear explanation of diapause in the introduction, focusing on its relevance to the study.

- It would be beneficial to present the advantages and necessity of integrating transcriptomics and metabolomics in molecular mechanism studies.

[Method]

- It is necessary to specify the number of samples used in the transcriptomics and metabolomics analyses.

- I can’t determine the statistical analysis been performed appropriately and rigorously. It is necessary to specify the data analysis methods, statistical method, and the statistical software used in the transcriptome and metabolome analysis.

- The sample preparation method for metabolomics, as well as the LC-MS analysis procedure, should be described.

- It is necessary to present information on the identified metabolites in a supplementary table with retention time, m/z, ppm error, CV%, intensity or fold change between diapause and pre-diapause stages, p-value.

[Results]

- 9(S)-HODE and PA(8:0/a-15:0) are not commonly observed under metabolite and lipid profiling conditions. Have their peaks been confirmed, and have they been validated using standard compounds?

- In Figure S4, the colors and shapes of PCR and RNA-seq data are too similar, making them difficult to distinguish. It is necessary to represent them differently.

- The content of the following sentence does not match the figure legend, and the labels "A" and "B" are missing from Figure 3. Revisions are needed.

“50 of 107 were higher in PD (Figure 3B), and the top 5 up-regulated metabolites were Adenylosuccinate, 9(S)-HODE, PA(8:0/a15:0), N6-(1,2-Dicarboxyethyl)-AMP, and N-acetylaspartate. 57 of 107 were higher in D, and the top 5 down-regulated metabolites were 5-Hydroxy-L-tryptophan, PE (15:0/22:4(7Z,10Z,13Z,16Z)), Sphingosine, Pantothenic Acid, and Glycerophosphocholine.”

- It is difficult to find the level changes of DEGs in the paper through figures or tables. It is necessary to present the data more clearly for easier reference.

“These results showed that, in the developmental process from PD to D, most DEGs were related to the immune defense, energy metabolism and hormone synthesis. There were three down-regulated DEGs related to insect immunity and defense (heat shock protein genes (HSP), Superoxide Dismutase (SOD), and catalase(CAT)), six down regulated DEGs related to energy metabolism (hexokinase (HK), pyruvate kinase (PK), ATP-dependent 6-phosphofructokinase (PFK), citrate synthase (CS), isocitrate dehydrogenase [NADP] cytoplasmic (IDH), α,α-trehalose-phosphate synthase [UDP forming] (TPS)), and two down-regulated DEGs related to hormone synthesis (juvenile hormone epoxide hydrolase (JHEH) and 20-hydroxyecdysone (20E)).”

“In this study, it was found that JHEH was down-regulated during diapause, thereby inhibiting the breakdown of juvenile hormones and keeping the larvae in the larval state for a long time, which may be the reason why SPB maintains diapause with mature larvae.”

[Discussion]

- Please provide the evidence based on the results to support the conclusion stated in the sentence on page 17, line 253 of the Discussion section.

“it is of great significance to study the transcriptome and metabolome of soybean insect at different diapause stages.”

- If there are existing studies related to the following sentence, relevant references should be added.

“Certain metabolites, such as Phosphatidyl choline (PC) and proline, can act as cryoprotectants to improve cold tolerance in diapausing insects.”

- Figure 4 appears to represent the key metabolic pathways identified in this study. Therefore, it is necessary to provide an interpretation and discussion of the key metabolites and transcriptomic changes observed in the study based on the measured results. Including unverified metabolites and transcripts in the interpretation may lead to overstatements. For example, regarding glycerol aldehyde 3-phosphate (G3P), the discussion includes changes in glycerol kinase (GK), glucose, glycogen, and glycerol. It is important to specify whether changes in glycerol kinase (GK), glucose, glycogen, and glycerol were actually observed.

- The title "4.2 Immune Response in the Diapause" primarily refers to immune response; however, the content mainly discusses ROS and antioxidants. A more appropriate revision of the title is needed to accurately reflect the content.

[Minor]

- Spelling corrections are needed throughout various sections of the paper.

Reviewer #2: General comments:

This study investigates the molecular basis of diapause in Leguminivora glycinivorella, a major soybean pest responsible for significant economic losses. By integrating transcriptomic and metabolomic analyses, the authors compared gene expression and metabolite profiles between pre-diapause and diapause stages, identifying differentially expressed genes and significantly altered metabolites. The findings highlight three critical molecular events during diapause—energy reserve accumulation, immune enhancement, and hormonal regulation—which collectively serve as adaptive mechanisms for survival under stress conditions. These findings advance understanding of diapause regulation in L. glycinivorella.

Major comments:

1. All figures in the manuscript are of insufficient resolution, making it difficult to discern the details and interpret the data accurately.

2. The qPCR results are not adequately described in the main text, and their significance is unclear. Additionally, the genes IMP3, RIOK2, and RPP25L mentioned in Figure S4 and Table S1 are not introduced or discussed. Furthermore, there is a repetition in the listing of the RPP25L-F primer sequences, which needs to be corrected.

3. While KEGG pathway enrichment analysis identifies key pathways such as the TCA cycle and glycolysis, the study fails to explain how these pathways interact to regulate diapause. For instance, it remains unclear whether energy conservation during diapause is achieved through suppressed glycolysis or enhanced lipid metabolism.

4. Some of the bioinformatic tools, such as HISAT, StringTie, and DESeq2, lack reference citations.

6. PLOS authors have the option to publish the peer review history of their article (what does this mean? ). If published, this will include your full peer review and any attached files.

**Do you want your identity to be public for this peer review?** For information about this choice, including consent withdrawal, please see our Privacy Policy .

Reviewer #1: No

Reviewer #2: No

---

## [Author Response · Author response to Decision Letter 1]

14 Mar 2025

Responses to the reviewer’s comments

(Manuscript ID: PONE-D-24-57514)

Note: our responses to the reviewers are written in blue font, and changes to the manuscript are shown in red font. Line and page numbers refer to the “changes in red” version of the manuscript.

Reviewer1

The authors presented a study aimed at identifying the key substances of diapause regulation by differential expression genes and differential metabolites, using transcriptome and metabolomics in the diapause and pre-diapause stages in L. glycinivorella. They found that metabolic regulation involving TCA cycle, glycolysis, insulin signaling pathway, and glycerophospholipid metabolic pathway, which suggesting that they may be closely related to energy reserve, immune regulation and hormone regulation during diapause. I have a hard time finding key results related to the described discussion… Therefore, I recommend that some revisions be made and specific questions be addressed before the paper is published.

We also appreciate your clear and detailed feedback and hope that the explanation has fully addressed all of your concerns. In the remainder of this letter, we discuss each of your comments individually along with our corresponding responses.

1. [Abstract]�In the abstract, insulin signaling is presented as a major pathway; however, there is a lack of related discussion in the results and discussion sections.

Thanks for your suggestions. In the abstract, we believe that insulin signaling is presented as a major pathway, and we have made corresponding additions to this content in the results. Insulin signaling pathway was significantly enriched in transcriptome KEGG enrichment analysis (P<0.01), and we believe that insulin signaling pathway may be a major participant in diapause metabolite regulation, so this part of the content is supplemented as described below. In the discussion section, we discuss the serine/threonine protein kinase (AKT) along this pathway, as well as the related enzyme GYS, which may influence glycogen synthesis and metabolism.

Page 11, lines 241~252:

KEGG pathways of DEGs could give deeper insights into the physiological processes related to diapause. Of all gene with a KEGG annotation, 2262 up-regulated DEGs were assigned to 318 KEGG pathways, and 5065 down-regulated DEGs were assigned to 336 KEGG pathways. Up-regulated DEGs were classified into “Ribosome biogenesis in eukaryotes”, “RNA degradation”, “Nucleotide excision repair”, and “RNA polymerase” (Figure 1C). Down-regulated DEGs were classified into “Glycolysis/Gluconeogenesis”, “Amino sugar and nucleotide sugar metabolism”, “Peroxisome”, and “Tryptophan metabolism” (Figure 1D). Top 10 KEGG pathways were significantly enriched, are listed in Table S7. In addition, some metabolic pathways are also significantly enriched, such as insulin signaling pathway, which may be a major participant in the regulation of metabolites in diaphaesis (32), in which serine/threonine protein kinase (AKT) is involved.

Page 18, lines 402~423:

Glucose can also be converted into glycerol, which can be used as an anti-freezing protective agent to enhance the cold tolerance of insects (37). Glycerol kinase (GK) is a rate-limiting enzyme in glycerolipid metabolism by the conversion of glycerol aldehyde 3-phosphate (G3P) to phospholipids such as glyceryl phosphate. The catabolism of glycerol in vivo consists of a two-step reaction, in which glycerol is catalyzed by glycerol kinase and phosphorylated to produce glycerol 3-phosphate. Rapid cold acclimation of Sarcophaga crassipalpis and Spodoptera exigua showed that they acted as the main anti-freeze protective substances by increasing their glycerol content (47). During diapause, GK was significantly down-regulated, This indicates the accumulation of glycerol in SPB, suggesting that this enzyme is mainly used as a protective substance to enhance the cold resistance of insects to survive diapause environment. As a key core factor in insulin signaling pathway, PDK1 phosphorylates and activates serine/threonine protein kinase (AKT), thereby regulating glucose homeostasis (48). An important gene downstream of AKT is glycogen synthase kinase 3β (GSK3β), which is involved in glycogen biosynthesis and metabolism. GSK3β inhibits glycogen synthase by phosphorylating glycogen synthase (GYS). In this way, it inhibits glycogen biosynthesis (49). In this study, by inhibiting GSK3β, AKT was able to overcome this inhibition and increase GYS activity, thereby affecting glycogen content.In this study, no significant differences in metabolites such as glycogen, glucose and glycerol were detected, but significant changes in transcription factors were found in this process, conjecturing that the TCA cycle and glycolysis process are dynamic changes, The entire metabolic network is in homeostasis.

2. [Introduction]�It is necessary to present a concise and clear explanation of diapause in the introduction, focusing on its relevance to the study.

Thanks for your suggestions. On the basis of the original manuscript, we have added a concise explanation of diapause and deleted the characteristics of diapause, and finally added the correlation between diapause and the article. We have provided detailed explanations of the changes made in the revised manuscript, including the specific locations where these changes can be found.

Page 3, lines 58~81:

Diapause is defined as a period of suspended development in insects and other invertebrates during unfavorable environmental conditions (7). In insects, diapause is a means of avoiding mortality and adjusting to the harsh surroundings (8). Metabolic suppression is a diapause mark that enables insects to align arrange synchronize their life cycles to coincide with times that are suitable for development, growth, and reproduction and survive unfavorable seasons (9). Diapause is a developmental stop state that is genetically determined and can happen at the embryonic stage (10), larval stage (11), pupal stage (12) and adult stage (13). And the types of diapause Diapause types are divided into obligate and facultative according to whether it can be influenced by environmental factors (14). Insect diapause is a dynamic process successive phases, which includes three periods: pre-diapause, diapause and post-diapause. In the stage of pre-diapause, SPB stores food, changes behavior and reduces development when the stimuli reach some critical level (15). Diapause slows or even prevents SPB growth and development, mostly through decreased activity and feeding intake, as well as the inability to finish the pupate and eclosion stages of development. Insect diapause is characterized by changes in the body, primarily a decreased metabolic rate and an improved tolerance to temperature and moisture (16,17). When the conditions change to be suitable for SPB, the insects recover from diapause with the metabolic and the physiological level returning to normal (6). SPB is a typical univolitine univoltine insect with obligatory diapause, and the larvae enter the pods to feed and develop until they are fourth instar larvae. They leave the pods, and overwinter in cocoons in the soil layer of 3~ – 15 cm (18–20). In order to achieve the goal of controlling the reproduction of this pest in a more environmentally friendly way, it is necessary to understand the molecular mechanism of diapause in SPB.

3. It would be beneficial to present the advantages and necessity of integrating transcriptomics and metabolomics in molecular mechanism studies.

Thanks for your suggestions. The advantages and necessity of combining transcriptomics and metabolomics in the study of molecular mechanisms have been added to the original manuscript, as described below.

Page 5, lines 101~113:

Metabolomics is the subject of qualitative and quantitative analysis of small molecules simultaneously, mainly studying the exposure of living organisms Changes of metabolites in vivo after boundary stimulation may change regularly with time (15). Transcriptomics (RNA-Seq) is an important method to study gene function and structure. Studying various genes in individuals, tissues or cells under different conditions is an important aspect of genomics, which has short sequencing time, large sequencing capacity and low sequencing cost(16,17). The combination of the transcriptome and metabolome better reflects the environment of the cell. The aims of this research are analyzing differentially expressed genes and metabolites and their functions of SPB in diapause and pre-diapause period with RNA-seq technology and targeted liquid chromatography mass spectrometry (LC-MS), and the molecular mechanism of diapause will be revealed and to elucidate the molecular mechanisms underlying diapause.

4. [Method]�It is necessary to specify the number of samples used in the transcriptomics and metabolomics analyses.

We were really sorry for our careless mistakes. Thank you for your reminder. The number of samples for transcriptomic and metabolomic analysis has been added to the manuscript.

Page 6, lines 116~123:

All samples were collected from the soybean fields in Xing’an League, Inner Mongolia (122°52'E, 46°49'N) on September, 2023. The forth instar larvae, pre-diapause (PD, were white and still feeding) and diapause (D, were red and had stopped feeding and developing for about 30d, Figure S1), were stored in -80°C. Each sample was divided into two equal parts for the transcriptomics and metabolomics. A total of 24 samples from 2 periods were collected, and each sample was divided into two equal parts for transcriptomics and metabolomics. (Three replicates per period for the transcriptome and nine replicates per period for the metabolome).

5. I can’t determine the statistical analysis been performed appropriately and rigorously. It is necessary to specify the data analysis methods, statistical method, and the statistical software used in the transcriptome and metabolome analysis.

Thanks for your suggestions, For strict statistical analysis, data analysis methods, statistical methods, and statistical software used in transcriptome and metabolome analysis are specified in the manuscript and marked in red for easy observation.

Page 6, lines 125~205:

Total RNA was isolated from the tissue specimens employing TRIzol® Reagent in strict adherence to the supplier's protocol. Subsequently, RNA integrity assessment was conducted using an Agilent 5300 Bioanalyser system, followed by spectrophotometric quantification with a NanoDrop ND-2000 instrument. At Shanghai Majorbio Bio-pharm Biotechnology Co., Ltd. (Shanghai, China), the RNA purification, reverse transcription, library building, and sequencing procedures were completed. To create the SPB RNA-seq transcriptome library, 1μg of total RNA and Illumina® Stranded mRNA Prep, Ligation (San Diego, CA) were used. The NovaSeq X Plus platform (PE150) was used to perform the sequencing library after Qubit 4.0 quantification, using the NovaSeq Reagent Kit (NovaSeq6000).The workflow comprised: PolyA+ mRNA enrichment via oligo(dT) magnetic bead selection followed by chemical fragmentation; First-strand cDNA synthesis with random hexamer primers using SuperScript™ ds-cDNA Synthesis Kit (Invitrogen, CA); Second-strand synthesis, end-repair/A-tailing, and Illumina adapter ligation per standardized workflows; Size selection (~300 bp fragments) through 2% Low Range Ultra Agarose electrophoresis; PCR amplification (15 cycles) with Phusion High-Fidelity DNA Polymerase (NEB, USA); Library quantification via Qubit® 4.0 Fluorometer prior to paired-end sequencing (PE150) on NovaSeq 6000 Plus platform with NovaSeq Reagent Kits. Clean reads in orientation mode were individually aligned to the reference genome using HISAT2 software(27). http://www.ncbi.nlm.nih.gov/datasets/genome/GCF_023078275.1/ is the website from which we obtained the SPB reference genome. Using a reference-based process, StringTie put together each sample's mapped reads(28). Each transcript's expression level was determined using the transcripts per million reads (TPM) method. The analysis of differential expression was done with DESeq2(29). Genes with FDR="0.05" and |log2FC|≧1 were identified as substantially divergent expressed genes (DEGs)(30). GO and KEGG functional-enrichment analysis were carried out by Goatools and Python scipy software, respectively, were performed to identify DEGs that were significantly enriched in GO and metabolic pathways.

2.3 Extraction and analysis of Metabolite

50 mg SPB were transferred into 2 mL cryotubes preloaded with 6 mm stainless steel grinding beads. Metabolite extraction was initiated by adding 400 μL of ice-cold methanol:water (4:1, v/v) solution containing 0.02 mg/mL L-2-chlorophenylalanine (internal standard). Mechanical homogenization was performed using a Wonbio-96c cryogenic grinder (Shanghai Wanbo Biotechnology Co., Ltd.) at 50 Hz for 6 minutes under -10°C conditions, immediately succeeded by ultrasonication at 40 kHz for 30 minutes in a temperature-controlled chamber (5°C). Following this, the homogenates underwent: ① Cryopreservation at -20°C for 30 minutes to enhance macromolecule precipitation; ② Phase separation via centrifugation at 13,000 × g for 15 minutes (4°C); ③ Precise supernatant collection using calibrated micropipettes; ④ Final filtration through 0.22 μm into LC-MS certified vials prior to instrumental analysis.

Using an ACQUITY HSS T3 column (100 mm × 2.1 mm i.d., 1.8 μm; Waters, USA), The SPBs were subjected to LC-MS/MS analysis at Majorbio Bio-Pharm Technology Co. Ltd. (Shanghai, China) using a UHPLC-Q Exactive HF-X system, equipped with an ACQUITY HSS T3 column (100 mm × 2.1 mm i.d., 1.8 μm; Waters, USA). The mobile phases consisted of 0.1% formic acid in water:acetonitrile (95:5, v/v) (solvent A) and 0.1% formic acid in acetonitrile:isopropanol:water (47.5:47.5, v/v) (solvent B). The flow rate was 0.40 mL/min and the column temperature was 40℃. The injection volume was 3 μL.

Thermo UHPLC-Q Exactive HF-X Mass Spectrometer was the mass spectrometer used for data collection. There are two possible modes of operation for its electrospray ionization (ESI) source. The conditions were set as followed: Aux gas heating temperature at 425℃; Capillary temp at 325℃; sheath gas flow rate at 50 psi; Aux gas flow rate at 13 psi; ion-spray voltage floating (ISVF) at -3500V in negative mode and 3500V in positive mode, respectively; Normalized collision energy , 20-40-60 eV rolling for MS/MS. Full MS resolution was 60000, and MS/MS resolution was 7500. Data acquisition was performed with the Data Dependent Acquisition (DDA) mode. The detection was carried out over a mass range of 70-1050 m/z. A uniform format was created from the UHPLC-MS raw data using Progenesis QI software (Waters, Milford, USA) through baseline filtering, peak identification, peak integral, retention time correction, and peak alignment. Then, the data matrix containing sample names, m/z, retention time and peak intensities was exported for further analyses (TableS1). Majorbio Biotechnology Co., Ltd. (Shanghai, China) self-compiled the Majorbio Database (MJDB), the HMDB (http://www.hmdb.ca/), and Metlin (https://metlin.scripps.edu/) as the primary databases for metabolite identification. Using the R package "ropls" (Version 1.6.2), the discriminant principle component analysis (PCA), least partial squares discriminant analysis (PLS-DA), and a seven-cycle interactive validation were performed(31). They evaluated the model's stability. Version 1.6.2 of the PLS-DA model's variable importance projection (VIP) was employed to check for differentially accumulating metabolites(32). Differentially accumulated metabolites (DAMs) were defined as those in which the VIP was greater than 1 and the Fold Change was greater than 2 or less than 0.5. Two groups' distinct metabolites were mapped into their corresponding biochemical pathways using metabolic enrichment and pathway analysis based on the KEGG database (http://www.genome.jp/kegg/).

2.4 Combined Analysis of the Transcriptome and Metabolome of SPB

Using internet resources, metabolome and transcriptome data were integrated using OmicsPLS (R packages) software for bidirectional orthogonal projections to latent structures (O2PLS). We were able to extract information about sha

---

## [Editor Report · Decision Letter 1]

19 Mar 2025

Insights from the transcriptome and metabolome into the molecular basis of diapause in Leguminivora glycinivorella (Lepidoptera, Olethreutidae)

PONE-D-24-57514R1

Dear Dr. XUE,

We’re pleased to inform you that your manuscript has been judged scientifically suitable for publication and will be formally accepted for publication once it meets all outstanding technical requirements.

Kind regards,

Munir Ahmad, PhD

Academic Editor

PLOS ONE
---

## [Editor Report · Acceptance letter]

PONE-D-24-57514R1

PLOS ONE

Dear Dr. XUE,

I'm pleased to inform you that your manuscript has been deemed suitable for publication in PLOS ONE. Congratulations! Your manuscript is now being handed over to our production team.

Kind regards,

on behalf of

Dr. Munir Ahmad

Academic Editor

PLOS ONE